

# Deep convolutional neural network architecture for facial emotion recognition

Dayananda Pruthviraja[1], Ujjwal Mohan Kumar[2], Sunil Parameswaran[2], Vemulapalli Guna Chowdary[2] and Varun Bharadwaj[2]

[1] Information Technology, Manipal Insitute of Technology, Manipal Academy of Higher Education, Bengaluru, Karnataka, India
[2] Department of Computer Science and Engineering, PES University, Bengaluru, Karnataka, India

## ABSTRACT

Facial emotion detection is crucial in affective computing, with applications in human-computer interaction, psychological research, and sentiment analysis. This study explores how deep convolutional neural networks (DCNNs) can enhance the accuracy and reliability of facial emotion detection by focusing on the extraction of detailed facial features and robust training techniques. Our proposed DCNN architecture uses its multi-layered design to automatically extract detailed facial features. By combining convolutional and pooling layers, the model effectively captures both subtle facial details and higher-level emotional patterns. Extensive testing on the benchmark Fer2013Plus dataset shows that our DCNN model outperforms traditional methods, achieving high accuracy in recognizing a variety of emotions. Additionally, we explore transfer learning techniques, showing that pre-trained DCNNs can effectively handle specific emotion recognition tasks even with limited labeled data. Our research focuses on improving the accuracy of emotion detection, demonstrating the model's capability to capture emotion-related facial cues through detailed feature extraction. Ultimately, this work advances facial emotion detection, with significant applications in various human-centric technological fields.

## INTRODUCTION

Facial emotion detection stands as a transformative breakthrough that not only bridges human-machine gaps but also revolutionizes decision-making and unravels human behavior intricacies. In today's landscape, integrating facial emotion detection is not merely additive but a force reshaping interactions. In the age of automation and artificial intelligence, machines' capacity to comprehend and reciprocate emotions gains pivotal importance. This technology introduces empathy to automated systems, altering communication, learning, and engagement dynamics.

Corresponding author
Dayananda Pruthviraja,
dayananda.p@manipal.edu

Use cases:

- Virtual learning: Enhancing virtual learning environments by adapting content based on students' emotional responses.
- Personalized e-commerce: Creating personalized shopping experiences by understanding customer emotions.
- Medical diagnostics: Improving medical diagnoses through emotional assessments, particularly in mental health.
- Social robotics: Developing more empathetic and interactive robots that can better understand and respond to human emotions.

Interpreting emotions from facial expressions faces challenges due to cultural nuances, context-based interpretations, and individual diversity. Ethical concerns involving privacy, consent, and data misuse spotlight its nuanced landscape. These challenges stimulate diverse hypotheses and perspectives, emphasizing the need for comprehensive field understanding.

Several open-source pre-trained models exist for facial emotion recognition, such as OpenFace (*Amos, Ludwiczuk & Satyanarayanan, 2016*), VGG-Net (*Simonyan & Zisserman, 2014*), Efficient Net (*Tan & Le, 2019*), and ResNet (*He et al., 2015*). These models, trained on large datasets like Fer2013 and Fer2013Plus (*Zahara et al., 2020*), extract facial emotion features. The goal is to improve how machines recognize these emotions, making human–computer interactions more natural and helpful. This can be achieved using powerful tools in deep learning like deep convolutional neural networks (DCNNs) to better understand the emotions people show on their faces. Our proposed DCNN model provides better facial detection accuracy compared to previously trained models. It offers faster model convergence due to a well-tuned learning rate scheduler and larger batch sizes, which allows it to process more data in each epoch.

## Motivation

The driving force behind the progression of facial emotion detection using DCNN algorithms stems from the profound significance of deciphering human emotional expressions. Emotions are intrinsic to human communication, and accurate and instant recognition could revolutionize human–computer interfaces, psychological assessments, and even the realm of social robotics. Extracting intricate emotional nuances from facial cuts could reshape technology into a more empathetic and adaptable entity, enhancing our interactions in the digital realm.

## Contribution

Our significant contribution lies in markedly enhancing the accuracy and classification capabilities of facial emotion detection through the implementation of DCNN algorithms. By devising innovative model architectures and employing advanced deep learning techniques, we have been able to capture intricate nuances of facial expressions that were previously challenging to discern. This has led to a substantial increase in accuracy rates, thereby enabling more precise emotion recognition. Moreover, our model's adeptness at classifying emotions, even in cases of subtle or complex expressions, sets a new standard for performance. Our research addresses the critical need for accurate and reliable emotion

classification, fostering a deeper understanding of human emotions and paving the way for applications that demand high-level emotional analysis, such as mental health diagnostics and human–computer interaction.

## MATERIALS AND METHODS

### Environmental setup

The experimental setup for this research was meticulously crafted to meet the demanding computational requirements of training and evaluating DCNN architectures tailored for facial emotion recognition. The hardware configuration boasted a powerful Intel Core i5 11th gen processor, augmented by a robust ensemble of graphics processing units (GPUs) including the Nvidia RTX 3050, alongside commercially available T4, A100, and V100 models, ensuring accelerated training capabilities. With ample memory (RAM) exceeding 4GB VRAM and high-capacity storage devices, the setup provided ample resources to seamlessly handle the intricacies of model development. On the software side, an operating system optimized for deep learning tasks was coupled with TensorFlow 2.9.0 for initial experimentation, transitioning to TensorFlow 2.15.1 for the latest reproducibility standards. GPU support was harnessed to maximize the potential of the deep learning framework, while essential Python libraries such as NumPy, Pandas, and Matplotlib facilitated streamlined data preprocessing, model evaluation, and visualization. The chosen development environment was meticulously curated to foster a seamless workflow, ensuring maximum productivity throughout the research process. The Fer2013Plus dataset, renowned for its comprehensive representation of facial emotions, served as the cornerstone for training and evaluating the facial emotion recognition models. Hyperparameters such as learning rate, batch size, and optimizer were meticulously tuned to optimize model performance. Subsequent evaluation of the trained DCNN architecture encompassed a thorough assessment utilizing standard evaluation metrics including accuracy, precision, recall, and F1-score, culminating in a comprehensive understanding of its efficacy in recognizing facial emotions.

### Data pre-processing
#### Resizing and standardization

Each individual model is capable of adopting a unique target size for its input tensor. The size 48×48 is chosen to enhance the versatility and obtain a substantial amount of information. Additionally, the pixels were normalized by dividing the pixel values by 255 (*Gal & Rubinfeld, 2018*). By adjusting each pixel's range to span from 0 to 1 the model's convergence rate is accelerated. This strategic normalization step aligns with the objective of optimizing the training process.

#### Colour space conversion

The Fer2013Plus dataset contains Y'UV images. However, when they are loaded into the model, it is loaded as a three channel tensor (red, green, blue). These tensors are then converted into grayscale as it helps to better form relations between the images (*Chavolla et al., 2018*). It reduces complexity, reduces dimensionality(the images only have a single

color channel), reduces noise, and increases computational efficiency as they take up less space in the memory.

### Label encoding

The labels of the dataset have been loaded into the model as a one-hot encoded tensor. Since, one-hot encoding doesn't assume any magnitude, which basic integer values do, the model doesn't have any assumptions on the values of ones and zeroes. Additionally, under normal encoding circumstances the model has a higher probability of assuming a relation between the encoded integer values, which is mitigated in one-hot encoding.

### Data augmentation

To prevent overlearning, increase variation and for better generalization, the training images have been picked at random and have been flipped horizontally. It was determined that this level of augmentation is optimal and further augmentation leads to loss in semantics which can potentially cause unrealistic samples, resulting in the loss of original information.

## Model architecture

### Input layer

Input layer is used to define the tensor shape the model takes in as input. This layer processes the inputs which are 2D arrays of pixel values. It converts the arrays into 1D tensors, by multiplying the input shapes to get single values that are compatible with all the other convolutional layers in the model.

### Convolutional layer

Convolutional layers are the building blocks of a CNN model. They extract features and distinguish between images. They have kernels with a specified size that slide with a predefined stride value over the image pixel and obtain a dot product upon multiplication (*Albawi, Mohammed & Al-Zawi, 2017*). This produces a feature map which is unique to the filter. As an input to the next layer, the previous layers feature maps are added using a bias to get the input for the next convolution neuron. Initializing the weights of a neural network is crucial because it can influence the convergence speed, optimization landscape, and overall generalization of the model. Poor initialization can lead to slow convergence, vanishing gradients, and hinders the network's ability to learn meaningful features. The kernel initializer is a parameter in neural network layers that determines how the weights of the layer should be initialized at the beginning of training. It specifies a method for setting initial values for the weights in order to help the network converge more effectively and potentially improve its performance (*Xu & Wang, 2022*). A kernel initializer "He Normal"(used when exponential linear unit activation functions are used) is used in this model.

### Activation function

In the convolutional layers, the ELU activation function is used. It doesn't have the "dying relu problem" (*Lu, 2020*) which means the gradients for the remaining neurons become very small hindering further learning. To solve this problem ELU allows a small negative

slope for negative inputs. The ELU activation function can be expressed:

$$f(x) = \{x \leftrightarrow x > 0 \, a(e^x - 1) \text{otherwise}} \quad \text{(\textit{Clevert, Unterthiner \& Hochreiter, 2016})}. \quad (1)$$

Equation (1) represents a piecewise function that operates differently for positive and non-positive values of $x$. When x is greater than 0, $f(x)$ returns the input $x$ itself. For non-positive $x$ values, it computes a value using the formula $a(e^x - 1)$, where 'a' is a constant and 'e' is the base of the natural logarithm.

At the output layer of the model the Softmax activation function (*Nwankpa et al., 2018*) is used since it is a multiclass classification.

$$s(x_i) = \frac{e^{x_i}}{\sum_{j=1}^{n} e^{x_j}} \quad \text{(\textit{Abadi et al., 2015})}. \quad (2)$$

Equation (2) defines a softmax function $s(x_i)$ applied to a vector 'x' with 'n' elements. It calculates the exponential of each element $x_j$ in the vector, sums up all the exponential values, and then divides each individual exponential value by the sum to normalize the vector elements into probabilities.

It is designed to convert raw scores, or logits into probability distributions over multiple classes. During training the softmax function's output probabilities are compared to the label's using the loss function (categorical crossentropy; *Bessel & Bradley, 1818*) to update the model's parameters and improve its predictions.

### Normalization layer

A normalization layer is placed after a convolutional layer. During the forward pass in training, this layer takes in the feature map (output) of the previous layer as its input and gives the normalized version of it as an output for the next layer. It normalizes the data by calculating the mean and standard deviation of the feature map (mini-batch) of the previous layer and it subtracts the input by the mean and divides the difference by the standard deviation.

$$\mu_i = \frac{1}{K} \sum_{k=1}^{K} (x_{i,k})^2 \, (\textit{Thompson \& Wesolowski, 2018}) \quad (3)$$

$$\sigma_i^2 = \frac{1}{K} \sum_{k=1}^{K} (x_{i,k} - \mu_i)^2 \, (\textit{Zhang \& Sabuncu, 2018}) \quad (4)$$

$$\hat{x_i} = \frac{x_i - \mu_i}{\sqrt{\sigma_i^2 + \epsilon}} \quad (5)$$

$$y_i = \gamma \cdot \hat{x_i} + \beta. \quad (6)$$

Equation (3) calculates the mean $\mu_i$ of values of a vector $x_{i,k}$ across a set of K elements. It sums each element's value in $x_{i,k}$ for all K instances and then divides the sum by K to obtain the average value of the vector.

Equation (4) measures the variance $\sigma_i^2$ of a vector $x_i$ across a batch by calculating the average squared deviation of each value from the batch mean $\mu_i$. It sums the squared differences for all K instances and divides by K, indicating the overall variability of the data around the mean.

Equation (5) calculates the normalized value $x_i$ of an input $x_{i,k}$ by subtracting the batch mean and dividing by the batch standard deviation $\sqrt{\sigma_i^2 + \epsilon}$, ensuring the input has a mean of 0 and a standard deviation of 1.

Equation (6) transforms the normalized value of $\hat{x_i}$ by scaling with the parameter and shifting with the parameter $\beta$, resulting in the final output $y_i$. This step provides flexibility to the model, allowing it to adjust the normalized data during training.

This implies that at the output there is a zero mean and unit deviation, this results in lesser sensitivity to the initialized kernels weights. Crucially, it reduces the variance to zero and hence eliminates internal covariate shift in the model, which leads to lesser chances of vanishing gradients. It gives a regularized data which adds noise into the model which may help generalization, but lengthens training time. Overall, this is necessary due its advantage of eliminating internal covariate shift (*Ioffe & Szegedy, 2015*).

### Pooling layer

Pooling layers are generally placed before a convolutional layer in the model. It systematically reduces the dimensionality of the feature map of the previous layer using the hyperparameters specified like stride and pooling window. By doing so, it helps the model retain only the prominent features by selecting the maximum value which has the most significant activation (*Nirthika et al., 2022*). This gives an added benefit that unnecessary noise is not retained helping the model train on the most prominent features, this in turn increases generalization as it learns the model can now identify the features in different positions in the image. This can also help increase the depth of the model as it reduces the number of parameters to be trained for the layers after it, making it possible to add more convolutional layers with greater neuron count. As the model is able to generalize more and it can identify prominent features at differing positions and orientations, it helps prevent overfitting on the training data.

### Dropout layer

The dropout layer is a regularization technique that temporarily deactivates a randomly chosen fraction of neurons when training to prevent overfitting. It introduces a form of model averaging and a certain amount of noise in the network which encourages the model to learn more robust features. This layer also allows for training different subsets of neurons at different iterations of training, helping to improve generalization. While a dropout layer may slow down the convergence, it results in a more accurate model. The dropout layer has a hyperparameter known as dropout rate which is used to specify the number of neurons to be deactivated at the training iteration (*Park & Kwak, 2017*).

Table 1 gives a comprehensive description of the proposed DCNN model depicting the hierarchical arrangement of the layers along with the crucial information about the number of parameters the model is training on and the dimension of the images at each layer.

**Algorithm1: Proposed Model**

Step 1: Input the facial images from the Fer2013Plus dataset

Step 2: Pre-process the images by resizing them to 48×48 and normalizing them

Step 3: Images are classified into Anger, Contempt, Disgust, Fear, Happiness, Neutral, Sadness and surprise

Step 4: The proposed DCNN model is built with 2,364,744 trainable parameters

Step 5: The proposed DCNN model effectively learns the facial features

Step 6: Making predictions on the test data and analyzing the performance metrics

**Table 1  Proposed DCNN model architecture.**

| Layer (type) | Output Shape | Param # |
|---|---|---|
| input (InputLayer) | [(None, 48, 48, 1)] | 0 |
| conv2d_1 (Conv2D) | (None, 48, 48, 64) | 640 |
| batchnorm_1 (BatchNormalization | (None, 48, 48, 64) | 256 |
| conv2d_2 (Conv2D) | (None, 48, 48, 64) | 73792 |
| batchnorm_2 (BatchNormalization) | (None, 48, 48, 64) | 256 |
| maxpool2d_1 (MaxPooling2D) | (None, 16, 16, 64) | 0 |
| dropout_1 (Dropout) | (None, 16, 16, 64) | 0 |
| conv2d_3 (Conv2D) | (None, 24, 24, 128) | 73856 |
| batchnorm_3 (BatchNormalization) | (None, 24, 24, 128) | 512 |
| conv2d_4 (Conv2D) | (None, 24, 24, 128) | 147584 |
| batchnorm_4 (BatchNormalization) | (None, 24, 24, 128) | 512 |
| maxpool2d_2 (MaxPooling2D) | (None, 12, 12, 128) | 0 |
| dropout_2 (Dropout) | (None, 12, 12, 128) | 0 |
| conv2d_5 (Conv2D) | (None, 8, 8, 256) | 295168 |
| batchnorm_5 (BatchNormalization) | (None, 8, 8, 256) | 1024 |
| conv2d_6 (Conv2D) | (None, 12, 12, 256) | 590080 |
| batchnorm_6 (BatchNormalization) | (None, 12, 12, 256) | 1024 |
| maxpool2d_3 (MaxPooling2D) | (None, 6, 6, 256) | 0 |
| dropout_3 (Dropout) | (None, 6, 6, 256) | 0 |
| flatten (Flatten) | (None, 9216) | 0 |
| dense1 (Dense) | (None, 128) | 1179776 |
| batchnorm_7 (BatchNormalization) | (None, 128) | 512 |
| dropout_4 (Dropout) | (None, 128) | 0 |
| out_layer (Dense) | (None, 8) | 1032 |

It is structured to acquire hierarchical patterns through sequences of convolutional and pooling layers, followed by fully connected layers and a class prediction output.

Essential components and design decisions include:

1. Input layer: The network starts with an input layer, shaping it to accept designated image data of shape (48,48,1).

2. Convolutional layers: The architecture features three sets of 2-dimensional convolutional layers, each with two successive convolutions. These layers extract

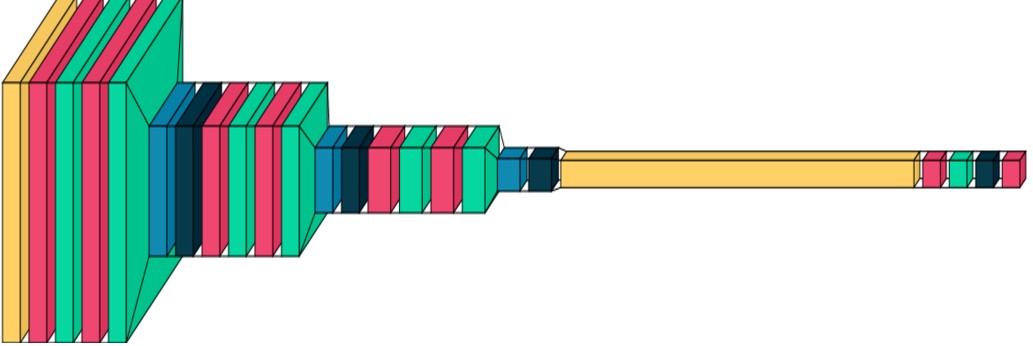

**Figure 1** **A graphical representation of the proposed DCNN model.** Plotted using visual keras (*Gavrikov, 2020*).

features through the exponential linear unit (ELU) activation function, mitigating gradient issues.

The first set (conv2d_1 and conv2d_2) employs 64 filters of size (3, 3) with 'same' padding. The second set (conv2d_3 and conv2d_4) enhances feature representation with 128 filters of size (3,3) with 'same' padding. The third set (conv2d_5 and conv2d_6) performs further feature extraction using 256 filters of size (3,3) with 'same' padding.

3. Batch normalisation layers: A batch normalization layer follows every convolutional layer and the final dense layer. Totaling to 7 batch normalization layers, they increase the model's training stability and increase convergence.

4. Max pooling layers: Max-pooling layers of pool size (2,2) follow each convolutional set (maxpool2d_1, maxpool2d_2, maxpool2d_3), minimizing feature map dimensions while conserving key information.

5. Dropout: Dropout layers with a dropout rate of 0.3 (dropout_1, dropout_2, dropout_3) and 0.4 (dropout_4) are integrated to mitigate overfitting by randomly deactivating neuron outputs during training.

6. Flatten layer: The ultimate max-pooled feature maps are compressed into a 1D vector, fed into fully connected layers.

7. Dense layers: Two fully connected layers are employed, with the initial dense layer (dense_1) encompassing 128 neurons and ELU activation.

8. Output layer: The final layer (out_layer) is a dense layer with neurons equivalent to the classification task's class count. It uses the softmax activation for class probabilities.

9. Model creation: The Keras functional API (*Lu, 2020*) amalgamates the entire structure, producing the ''DCNN'' model.

This architecture in Fig. 1 strives to balance feature extraction capacity and model complexity. Utilizing multiple convolutional layers, upping filter counts, and applying dropout and batch normalization, the model aims to acquire intricate hierarchical features from input images while averting overfitting. It's a suitable DCNN for image classification tasks—categorizing input images into distinct classes.

## Optimizers and learning rates

For training the proposed DCNN model, the Adam optimizer (*Kingma & Ba, 2014*), with an initial learning rate of 0.01, has been used. It is beneficial to use the Adam optimizer since it adjusts the learning rates of individual parameters based on the magnitude of historical gradients, and for faster learning, the optimizer accelerates gradient descent by taking an exponentially weighted average of the gradients. This optimizer also adjusts learning rate for each parameter by taking into account the recent gradient, resulting in speeding up convergence and reducing the need for manual learning rate tuning. This combination of adaptive learning rate and momentum helps reduce oscillations during optimization (*Kingma & Ba, 2017*).

Adam uses exponentially decaying moving averages of past gradients and squared gradients, estimating first and second moments without storing full gradient history, benefiting memory-efficient training, especially in deep networks.

## Model call backs

Two call back functions have been used during the training phase of the model.

### Early stopping

Early Stopping is a model call back technique in machine learning that halts the training process when a predefined performance metric ceases to improve or starts deteriorating (*Prechelt, 2000*). This call back has three main hyperparameters, monitor which has been set to monitor the validation loss, delta checks for the minimum change in the monitored metric, patience value is set to wait for a specified amount of epochs (a cycle through the training data) before the model training is stopped. This call back is mainly used to prevent overfitting of the model by stopping the model training process.

### Learning rate scheduler

A learning rate scheduler is a technique in machine learning that dynamically adjusts the learning rate during training. It helps optimize the training process by decreasing the learning rate over time, allowing the model to converge faster initially and then fine-tune as it approaches convergence. This aids in finding an optimal balance between rapid progress and stable refinement, resulting in better convergence and improved model performance (*Kim et al., 2021*). Specifically using Reduce LROnPlateau API from tensorflow keras callbacks (*Lu, 2020*) provides hyperparameters such as monitor, min_delta, mode, factor, patience, min_lr, *etc.* The hyperparameter mode is used to specify whether the learning rate should decrease when the quantity being monitored has stopped improving. Figure 2 shows the decrease in learning rate with epochs.

## RESULTS

The proposed DCNN model is run on different GPUs ranging from commercial grade machine learning optimized GPUs to gaming GPUs. It is done so to get an insight of how the model performs across GPUs with varying computational power and memory. It also helps estimate the scalability of the model for real-life deployment. By looking at the model's performance metrics, memory consumption and training time and other such key

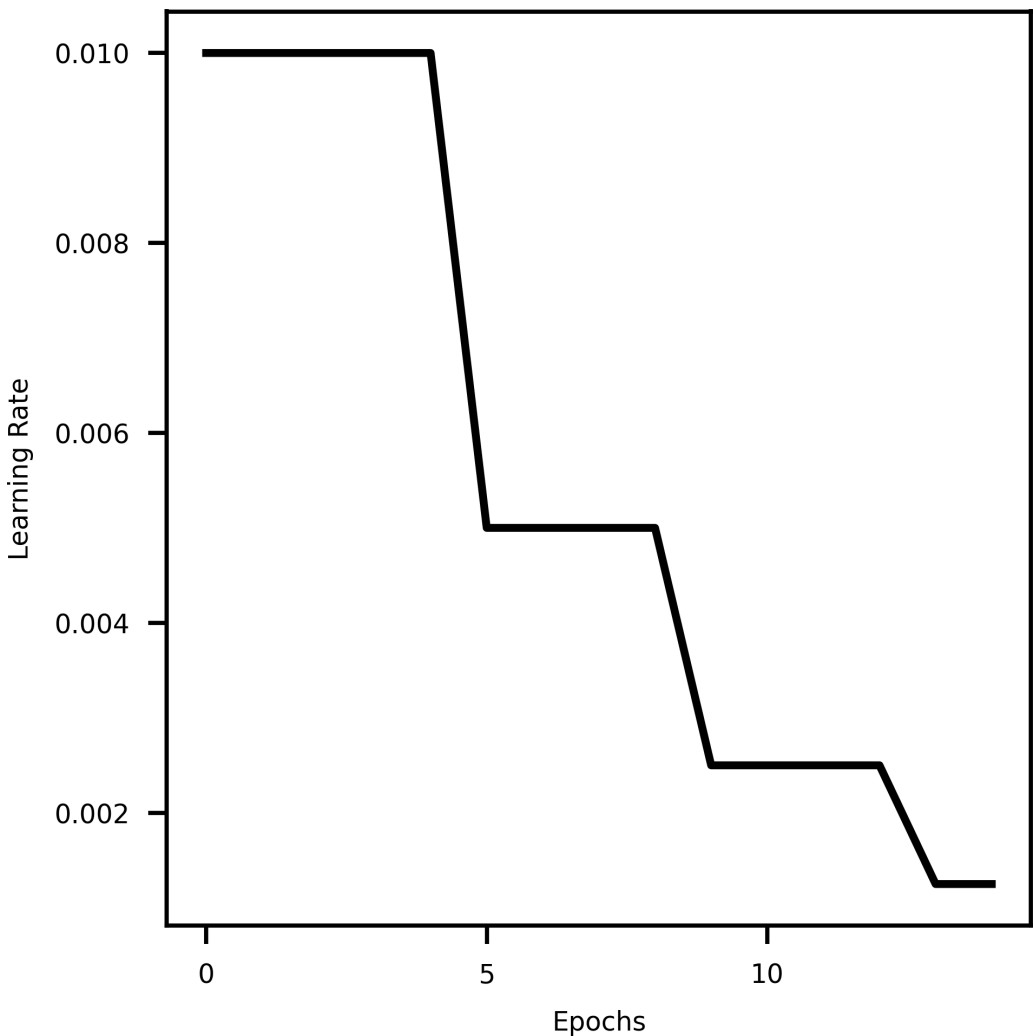

**Figure 2  A curve representing learning rate and epochs.**

parameters the model can be rated as to how consistent it is. The proposed model takes up about 2.4 GB of V-RAM when being trained on the FER2013Plus (*Zahara et al., 2020*) dataset. By doing so, an ideal infrastructure can be selected for the model which balances between training time and accuracy.

## Model accuracy

A graph of epochs *vs* Accuracy, depicted in Fig. 3, gives key insights on how the proposed model is learning on the data set. During model training for the first few epochs it is seen that the accuracy starts out low and increases quickly, which indicates the model is learning the basic features and patterns from the data. However, as the training progresses the curve tends to plateau which indicates that convergence is reached. Rapid changes in the curve indicates a higher learning rate which needs to be reduced in order to stop overshooting the optimal weights. The main purpose of this curve is to show overfitting when the model

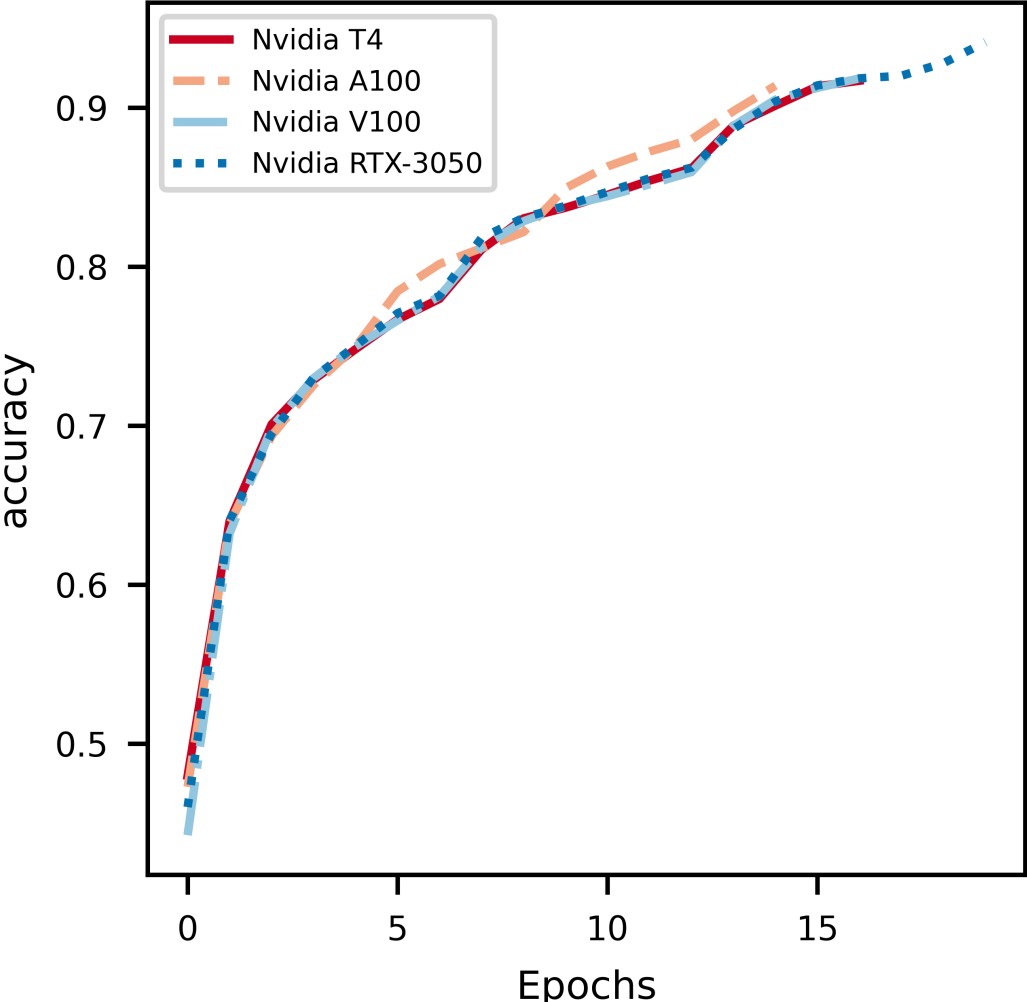

**Figure 3** An illustration of the change in accuracy with epochs.

learns too many patterns on the training data so it can't generalize it for the validation data set.

In Fig. 4, the epochs *vs* validation accuracy graph, a lot of fluctuations can be noticed which is due to the smaller size of the dataset used for evaluation. Notably, it can be seen that the validation accuracy increases, which supports the notion that the model is good at generalizing and not overfitting.

From the above graphs, we can inference that, early stopping is activated at different epochs for different GPUs. We see that the model trained on Nvidia T4 GPU has the best training accuracy (92.3%). The model reaches its best training accuracy by epoch 13 for the Nvidia RTX-3050 GPU this is because early stopping activates at a local minima during training. These effects can be further minimized by increasing the patience value.

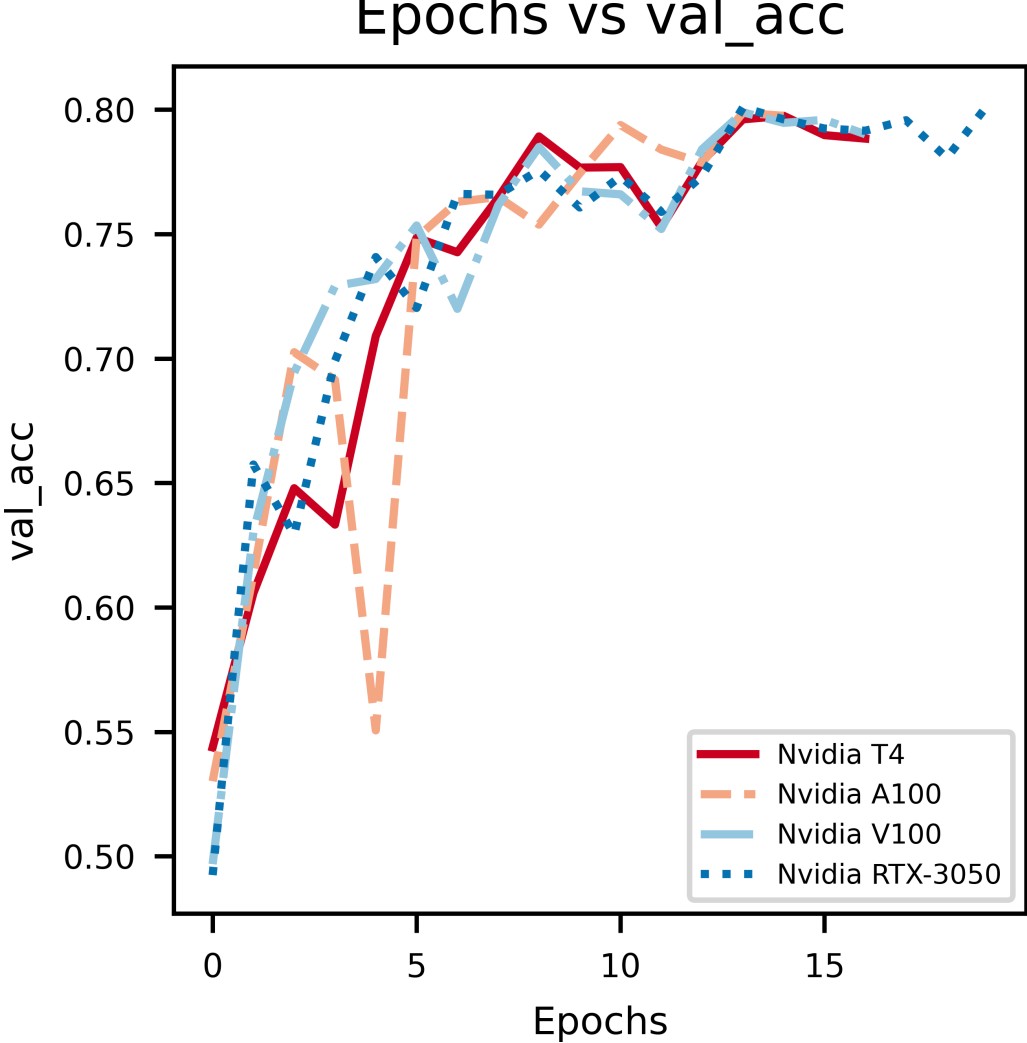

**Figure 4    An illustration of the increase in validation accuracy with epochs.**

The second figure displays epochs with respect to validation accuracy of inferencing the model with multiple GPUs separately. The Nvidia RTX-3050 GPU seems to have finished inferencing quickly in terms of epoch count at a cost of validation accuracy when compared to Nvidia T4, which takes longer to infer with respect to epoch count but provides better validation accuracy. Overall, this shows the versatility of the model as it can be trained on different kinds of GPUs with different V-RAM configurations and still achieve comparable results.

## Model loss
As it can be observed in Fig. 5, at the initial phase of training, the loss is high due to random weight initialization. However, in the subsequent epochs the loss reduces gradually hinting that the key features for each label has been learnt. The slope of the curve gradually decreases and then flattens, signifying that the model has converged. Further, to achieve

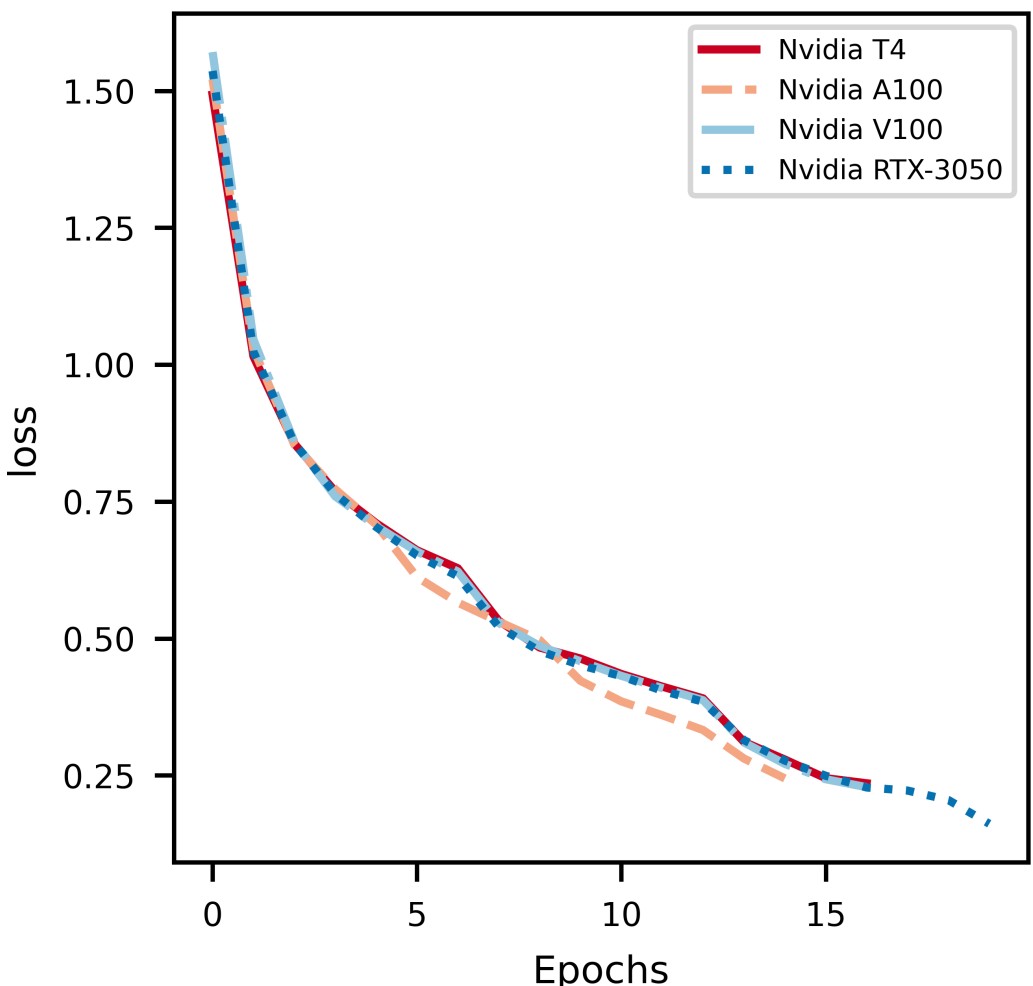

**Figure 5** **An illustration of the drop in loss with epochs.**

better loss values the model has to be fine tuned by training layers specifically. The learning rate scheduler callback decreases the learning rate with epochs, which further helps in decreasing the loss and preventing overfitting. The decrease in validation loss can be observed in Fig. 6. Comparing the validation loss and training loss graphs an insight into the bias–variance trade-off (*Yang et al., 2020*) of the model is obtained. Generally low training loss and high validation loss means overfitting whereas, high training loss and high validation loss means the model is underfitting. From Figs. 5 and 6, it is observed that the model has comparable training and validation losses leading to the conclusion that it is neither over-fitting nor under-fitting (*Zhang, Zhang & Jiang, 2019*).

The training loss *versus* epochs curve displays the effectiveness of the Learning Rate scheduler as this allows the model to reach loss convergence very fast across varying system conditions. It is observed that Nvidia A100 isn't the best case scenario for this model.

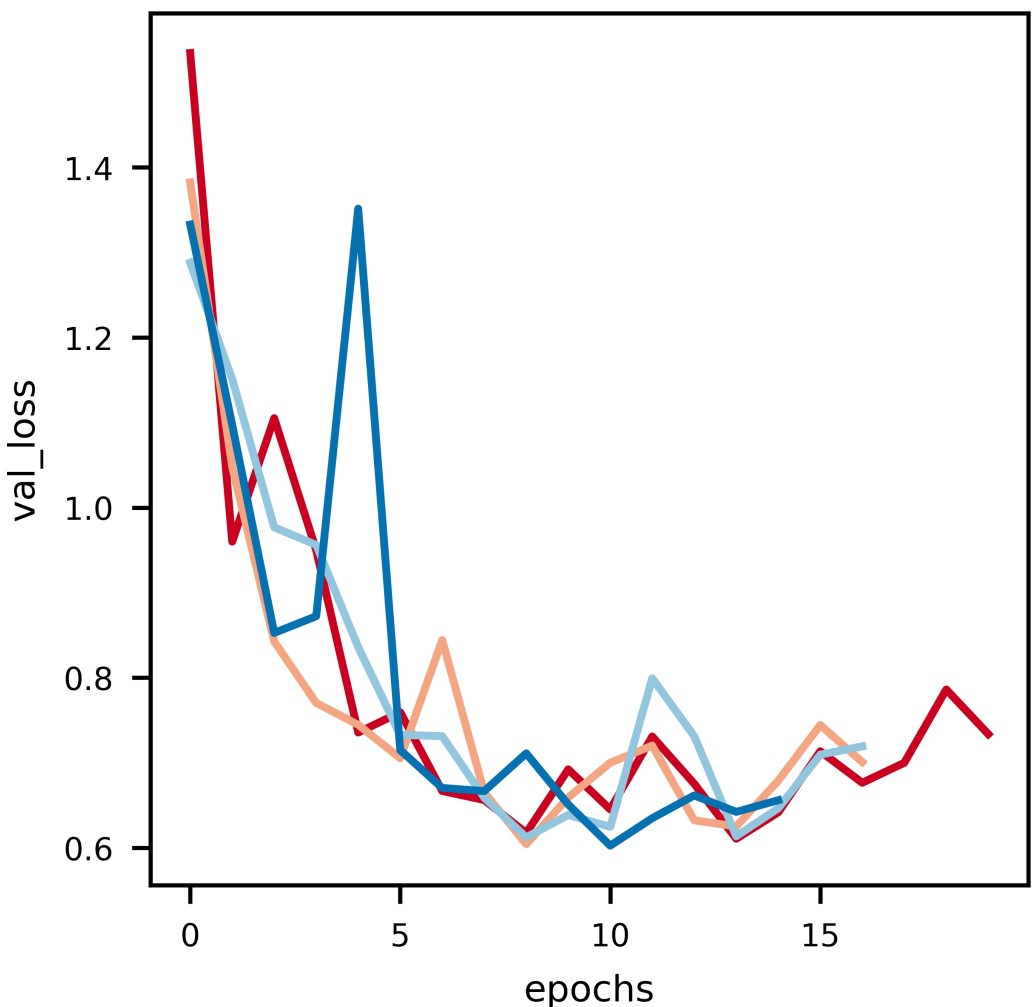

**Figure 6 An illustration of the drop in validation loss with epochs.** Noise is induced in this curve due to the small data size of validation data.

The validation loss *versus* epochs curve displays the versatility of validation loss during inference, Nvidia RTX-3050 GPU outperforms every other GPU when it comes to inferring as the validation loss is the least, every other GPU overlearns after a point hence leading to higher validation loss. This suggests that Nvidia RTX-3050 is better as generalising the dataset as it performs well during inferencing.

Figure 7 gives the training times taken by various GPUs to train the proposed DCNN model. It is found that the model trains faster on GPUs with more V-RAM.

The proposed DCNN model has been compared to other pre-trained models, namely, EfficientNetB4, VGG16, VGG19, ResNet50V2, on the Fer2013Plus dataset, and their performances were taken as a baseline. Figure 8 plots the differences between the proposed DCNN model with other models in terms of training accuracy, validation accuracy, training loss, validation loss. Table 2 gives a comprehensive overview of all the performance metrics

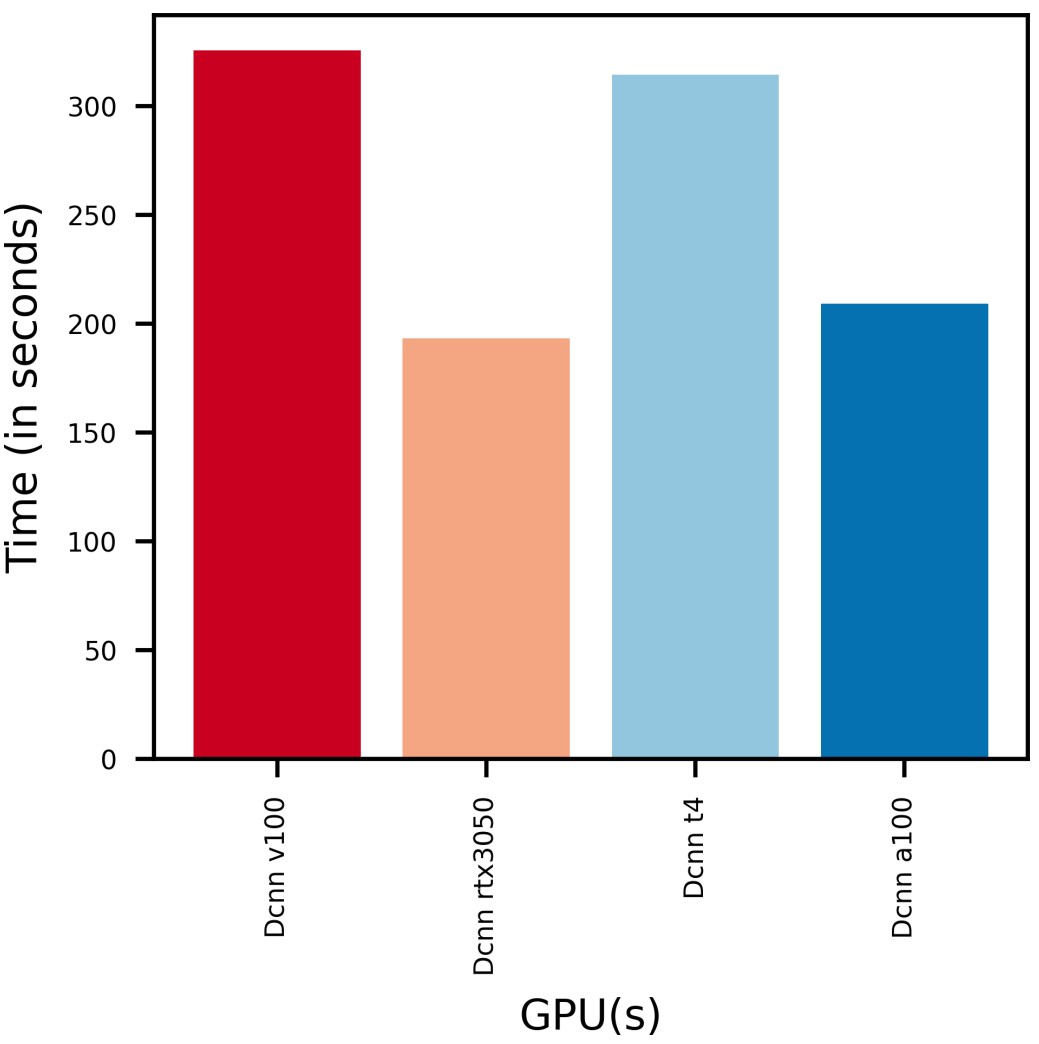

**Figure 7** **Training time of various GPUs.**

employed, namely validation accuracy, validation loss, training accuracy, training loss, F1-score, recall and precision.

### EfficientNetB4

EfficientNetB4 consisting of 475 layers within performs with around 36.5% of validation accuracy compared to 81.3% of validation accuracy on the proposed DCNN model. When observing the accuracy *vs* epochs curve a lot of noise and fluctuations are observed in the EfficientNetB4 model. This is due to the model being highly complex as it uses many layers and some of them are connected to each other non-sequentially using functional apis. Hence, the model tends to overlearn certain features and cannot generalize, which leads to it having the worst performance compared to any other pre-trained models that were compared.

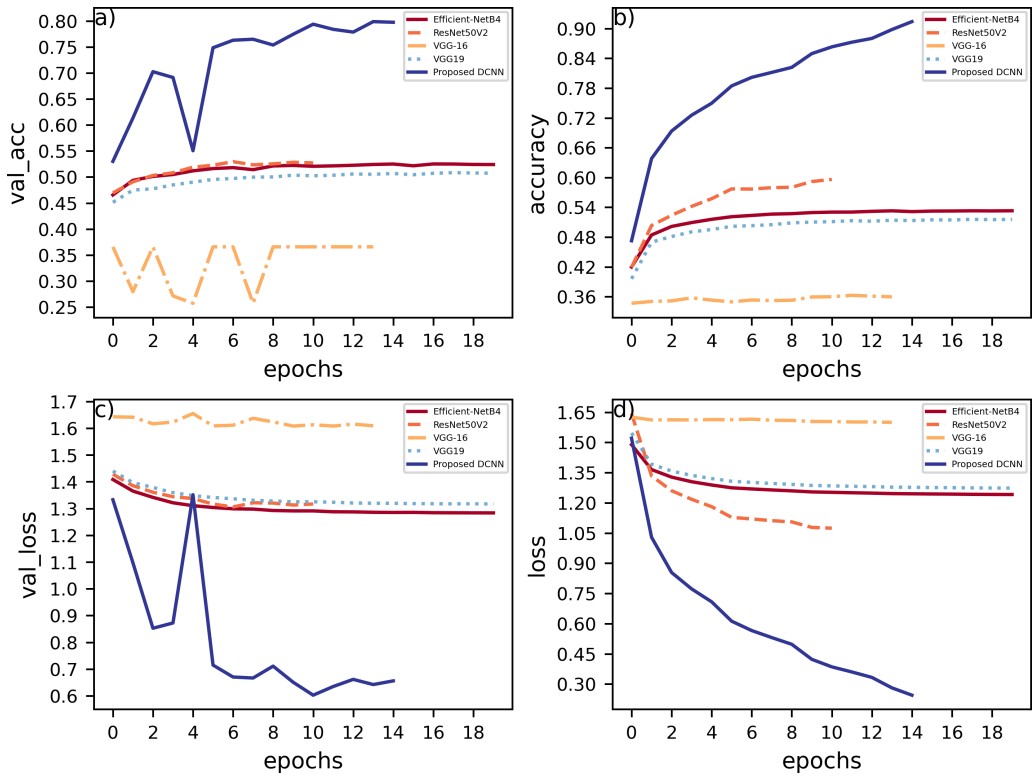

**Figure 8** (A) Training accuracies; (B) loss curves; (C) validation accuracies; (D) validation losses.

**Table 2** Comparison with other previously trained models with respect to the proposed DCNN model.

| Model Used | Val_Accuracy | Val_Loss | Accuracy | Loss | F1 score | Recall | Precision |
|---|---|---|---|---|---|---|---|
| Efficient-Net B4 | 36.583% | 1.607 | 36.314% | 1.5950 | 0.1960 | 0.3658 | 0.1330 |
| ResNet 50 | 52.444% | 1.282 | 53.410% | 1.2392 | 0.2472 | 0.2826 | 0.2290 |
| VGG 16 | 50.599% | 1.314 | 51.772% | 1.2706 | 0.2600 | 0.2949 | 0.2452 |
| VGG 19 | 45.231% | 1.478 | 45.850% | 1.4512 | 0.2520 | 0.2700 | 0.2405 |
| Proposed DCNN (Without Augmentation ) | 81.326% | 0.575 | 92.242% | 0.2280 | 0.2449 | 0.2511 | 0.2394 |
| Proposed DCNN (With Augmentation ) | 80.138% | 0.559 | 84.272% | 0.4401 | 0.2405 | 0.2498 | 0.2334 |

## VGG-19

The VGG-19 model has a layer count of 22 and has a validation accuracy of about 45.23%. This model mainly contains blocks of convolutional layers and pooling layers in them. This model is substantially better than EfficientNetB4 because the model architecture is simpler and the pooling layers reduces overlearning. This gave the idea to use blocks of convolution layers and max pool layers in the proposed DCNN model.

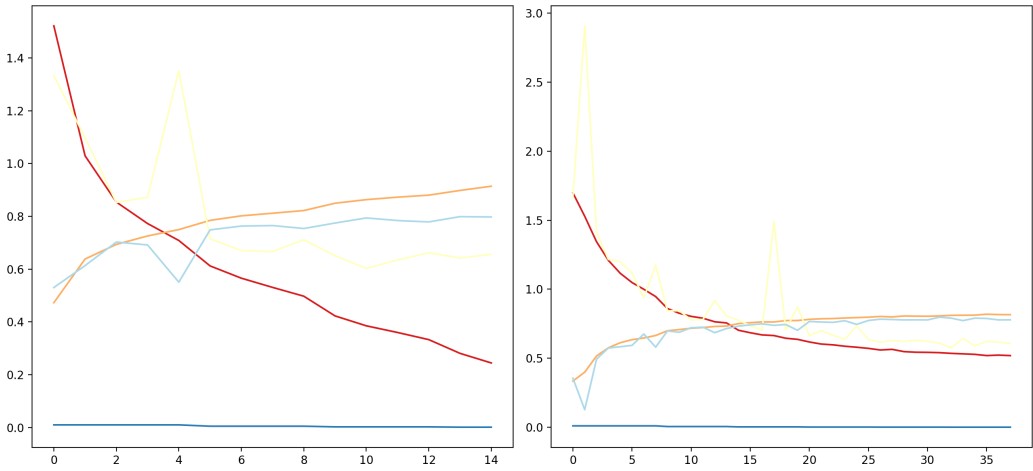

**Figure 9 Comparison of model performance.** (A) Without data augmentation; (B) with data augmentation.

## VGG-16

This model architecture is similar to that of VGG19 and performs slightly better compared to VGG19, it has a layer count of 19 and has a validation accuracy of 50.59%. This model is lighter in terms of the density of layers compared to all the other models as well as to the proposed DCNN model. Out of the compared models this is the simplest model that could achieve a comparatively good validation accuracy.

## ResNet50V2

This model has a layer count of 190 layers. This model focuses more on batch normalization and padding layers. This model performs the best as compared to the other pre-trained models discussed here. It has a validation accuracy of about 52.4%. This is a relatively complex model in terms of layer density compared to proposed DCNN.

Figure 9 can be used to infer that augmenting the data can lead to varied results and must be used cautiously.

## CONCLUSIONS

Concluding this analysis, the evaluation of the proposed DCNN model on the Fer2013Plus dataset showcases its robustness and adaptability through the strategic incorporation of convolutional, max pooling, and dropout layers. These architectural elements contribute significantly to the model's accuracy, striking a balance between overfitting and underfitting. The hierarchical combination of convolutional layers and pooling layers aids the model in extracting relevant features at different scales, ultimately contributing to its accuracy and generalization. The integration of dropout layers further acts as a regularizer, mitigating overfitting by preventing the model from relying excessively on specific neurons.

Through the careful orchestration of these layers, the DCNN model achieves a harmonious blend of feature extraction, dimensionality reduction, and regularization.

This approach ensures that the model learns pertinent and prominent features, avoids overfitting or underfitting, and attains impressive accuracy on the dataset.

## ACKNOWLEDGEMENTS

We thank IEEE Bengaluru chapter and MIT Bengaluru for providing us support to carry out this research.

### Funding
The authors received no funding for this work.

### Competing Interests
The authors declare there are no competing interests.

### Author Contributions
- Dayananda Pruthviraja conceived and designed the experiments, authored or reviewed drafts of the article, and approved the final draft.
- Ujjwal Mohan Kumar conceived and designed the experiments, performed the experiments, analyzed the data, performed the computation work, prepared figures and/or tables, authored or reviewed drafts of the article, and approved the final draft.
- Sunil Parameswaran performed the experiments, performed the computation work, prepared figures and/or tables, and approved the final draft.
- Vemulapalli Guna Chowdary analyzed the data, performed the computation work, authored or reviewed drafts of the article, and approved the final draft.
- Varun Bharadwaj analyzed the data, performed the computation work, prepared figures and/or tables, authored or reviewed drafts of the article, and approved the final draft.

### Data Availability
Data can be found in the Supplemental Files.

The dataset used to train the model, the dataset is available at GitHub: https://github.com/microsoft/FERPlus.git.

### Supplemental Information
Supplemental information for this article can be found online at http://dx.doi.org/10.7717/peerj-cs.2339#supplemental-information.

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
