# Peer review of "Deep convolutional neural network architecture for facial emotion recognition"

_PeerJ Computer Science, doi:10.7717/peerj-cs.2339_

## Round 0.1 · original submission · Major Revisions

Based on the referee reports, I recommend a major revision of the manuscript. The author should improve the manuscript, taking carefully into account the comments of the reviewers in the reports and resubmit the paper.

Reviewer 1 ·

Basic reporting

1. The authors do not provide the Python code and dependent packages. I suggest the authors provide this information to improve the reproducibility.
2. The resolution of the figures is very low. It is difficult to see the text in the figure clearly.
3. I suggest the authors re-organize the manuscript. The model comparison should occur in the method and result session rather than in the discussion session.

Experimental design

1. The authors do not clearly define the training process. They do not clarify the dataset split, including the training/validation/testing set.
2. It seems that the authors do not test on the testing set, so the result may be overestimated.
3. There is “transfer learning” In the title and abstract but I cannot find any implementation of transfer learning in the main text.
4. The author said that their model is interpretable but I cannot find any interpretation experiment in the main text.
5. The comparison across models are also not clear. The author does not mention any details of the comparison. I am not sure the author has run all of these models on the same condition.

Validity of the findings

The results are not rigorous and solid. The title and abstract do not match the main text. I suggest the authors make significant improvements on these issues.

Cite this review as

Reviewer 2 ·

Basic reporting

All comments have been added in detail to the 4th section called additional comments.

Experimental design

All comments have been added in detail to the 4th section called additional comments.

Validity of the findings

All comments have been added in detail to the 4th section called additional comments.

Additional comments

Review Report for PeerJ Computer Science
(Deep convolutional neural network architecture for facial emotion recognition using transfer learning)

1. Within the scope of the study, classification operations were carried out using the FER+ dataset, which was announced by Microsoft at an ACM conference in 2016 and shared as open source.

2. The dataset used was announced approximately 8 years ago and has received over 700 citations since its announcement, so it is a subject that has been studied extensively.

3. Although many studies have been done on the dataset, there are serious deficiencies in the literature review section of this study.

4. Various basic simple data preprocessing and data augmentation operations were performed on the dataset. In order to fully observe whether this process really has a positive effect on the results, it is necessary to compare the classification results before and after data preprocessing/augmentation in the dataset.

5. It is recommended to add sample images for each class in the dataset, both in the initial version and after data preprocessing/augmentation.

6. In the study, a very basic and simple model is proposed as a classification model. Since the dataset size used was very small, the network depth was accordingly not very deep.

7. More information such as the toolbox/framework used in the implementation phase of the study should be included. Also, how are hyperparameters such as optimizer, learning rate, epoch determined? Have different attempts been made? Please explain in detail.

8. The proposed simple model was compared with EfficientNet, VGG and ResNet models that are frequently used in the literature. Although there are more up-to-date state-of-the-art models that can be used in the literature, why were these models preferred for comparison, and why were more up-to-date models not used?

9. There are serious deficiencies in the evaluation metric stage. Important metrics such as confusion matrix, ROC curve, AUC score should be included. Cross-validation is very important for accurate analysis of the results.

As a result, there are serious question marks regarding the originality of the study, its main basic deficiencies are stated above.

Cite this review as

Reviewer 3 ·

Basic reporting

1. abstract needs a revision to highlight novelty.
2. The introduction section lacks proper research motivation, research questions/objectives .
3. The literature review is quite poor and needs major overhauling to justify the research gap being identified from recent similar works.

4.

Experimental design

1. The mathematical modeling is poor, needs proper revision.
2. The proposed algorithm is also quite unprofessional, needing a revision.

Validity of the findings

1. The results section needs to be arranged in terms of the posed addressed research questions.
2. Threats to external validity are not addressed.
3. How your approach contribute to technological applied areas.

Additional comments

paper needs through proofreading

Cite this review as

---

## Round 0.2 · Major Revisions

Kindly revise the manuscript as per the reviewer suggestions and resubmit it. The resubmission needs to include a detailed rebuttal letter for the *original* set of peer-review comments.

Reviewer 1 ·

Basic reporting

Some points have improved but the authors do not reply point by point.

Experimental design

Some points have improved but the authors do not reply point by point.

Validity of the findings

Some points have improved but the authors do not reply point by point.

Cite this review as

Reviewer 2 ·

Basic reporting

All comments have been added in detail to the last section.

Experimental design

All comments have been added in detail to the last section.

Validity of the findings

All comments have been added in detail to the last section.

Additional comments

Review Report for PeerJ Computer Science
(Deep Convolutional Neural Network Architecture for Facial Emotion Recognition)

Thanks for the revision. I have reviewed the revised paper in detail, but although there are some very limited improvements, there are still deficiencies. Also, I did not see any response to my first round reviewer comments consisting of 9 items in the response letter. Please respond to the first round reviewer comments in detail step by step, and then indicate which parts of the paper you were able to improve. Best regards.

Cite this review as

---

## Round 0.3 · Minor Revisions

Kindly revise the manuscript as per the reviewer1 suggestions and resubmit it.

Reviewer 1 ·

Basic reporting

Issues have been improved but some issues have not been solved. Please see additional comments.

Experimental design

Issues have been improved but some issues have not been solved. Please see additional comments.

Validity of the findings

Issues have been improved but some issues have not been solved. Please see additional comments.

Additional comments

Interpretation analysis needs to analyze the features contributing to prediction. The author mentions the interpretation analysis at the beginning of the paper but I find no interpretation analysis involved in the analysis. There is also no relevant result in the supplementary code provided by the author.

Cite this review as

Reviewer 2 ·

Basic reporting

All comments have been added in detail to the last section.

Experimental design

All comments have been added in detail to the last section.

Validity of the findings

All comments have been added in detail to the last section.

Additional comments

Review Report for PeerJ Computer Science
(Deep Convolutional Neural Network Architecture for Facial Emotion Recognition)

Thanks for the revision. Some responses are very limited, but the changes made to the paper after the revision are generally at a certain level. I recommend that the paper be accepted. I wish the authors success in their future projects. Best regards.

Cite this review as

---

## Round 0.4 · accepted · Accept

Author has addressed reviewer comments properly. Thus I recommend publication of the manuscript.

Reviewer 1 ·

Basic reporting

Please see additional comments

Experimental design

Please see additional comments

Validity of the findings

Please see additional comments

Additional comments

The issues in the manuscript have been improved by removing the descriptions of model interpretation.

Cite this review as